# ACE2 Receptor and Its Isoform Short-ACE2 Are Expressed on Human Spermatozoa

**DOI:** 10.3390/ijms23073694

**Published:** 2022-03-28

**Authors:** Marina Ramal-Sanchez, Chiara Castellini, Costanza Cimini, Angela Taraschi, Luca Valbonetti, Arcangelo Barbonetti, Nicola Bernabò, Barbara Barboni

**Affiliations:** 1Faculty of Biosciences and Technology for Food, Agriculture and Environment, University of Teramo, 64100 Teramo, Italy; mramalsanchez@unite.it (M.R.-S.); ccimini@unite.it (C.C.); ataraschi@unite.it (A.T.); lvalbonetti@unite.it (L.V.); bbarboni@unite.it (B.B.); 2Andrology Unit, Department of Clinical Medicine, Life, Health and Environmental Sciences, University of L’Aquila, 67100 L’Aquila, Italy; chiara.castellini@univaq.it (C.C.); arcangelo.barbonetti@univaq.it (A.B.); 3Experimental Zooprophylactic Institute of Abruzzo and Molise ‘G. Caporale’ (IZSAM), 64100 Teramo, Italy; 4Institute of Biochemistry and Cell Biology (CNR-IBBC/EMMA/Infrafrontier/IMPC), National Research Council, Monterotondo Scalo, 00015 Rome, Italy

**Keywords:** spermatozoa, ACE2, short-ACE2, male reproduction, fertility, SARS-CoV-2

## Abstract

Angiotensin-converting enzyme 2 (ACE2) is a protein widely expressed in numerous cell types, with different biological roles mainly related to the renin-angiotensin system. Recently, ACE2 has been in the spotlight due to its involvement in the SARS-CoV-2 entry into cells. There are no data available regarding the expression of ACE2 and its short-ACE2 isoform at the protein level on human spermatozoa. Here, protein expression was demonstrated by western blot and the percentage of sperm displaying surface ACE2 was assessed by flow cytometry. Immunocytochemistry assays showed that full-length ACE2 was mainly expressed in sperm midpiece, while short ACE2 was preferentially distributed on the equatorial and post-acrosomal region of the sperm head. To our knowledge, this is the first study demonstrating the expression of protein ACE2 on spermatozoa. Further studies are warranted to determine the role of ACE2 isoforms in male reproduction.

## 1. Introduction

Angiotensin-converting enzyme 2 (ACE2) was first discovered and cloned in 2000 as a homologue of ACE [1,2]. It is a membrane exopeptidase, a single-pass type I transmembrane protease enzyme (glycoprotein) constituted by 805 amino acids and is expressed by multiple organs and cell types, including lungs [3], kidneys, heart [4], testis [5] and alveolar epithelial cells and enterocytes from the small intestine [6]. ACE2 is a key component of the renin-angiotensin system (RAS) [7], which regulates both the blood pressure and fluid/electrolyte homeostasis [8]. It acts as a convertor of angiotensin I to angiotensin 1–9 (Ang-(1–9)) and the potent vasoconstrictor angiotensin II in the vasodilator peptide angiotensin 1–7 (Ang-(1–7)), conferring to the enzyme a cardioprotective role [7]. Moreover, ACE2 has been proposed as a modulating factor in the pathophysiology of diabetes mellitus [9] and renal failure [10], thus extending the role of the RAS system and the molecules involved well beyond the fluid balance and blood pressure control.

One of the most intriguing aspects of this protein is its dual role [11,12], acting as a protector and modulator of multiple pathophysiological processes but also implicated in the entry of viruses within the cells. The dual role of ACE2 has remained somehow in the background since December 2019, with the first reported cases of the virus named Severe Acute Respiratory Syndrome CoronaVirus-2 (SARS-CoV-2) responsible for the Coronavirus Disease 2019 (COVID-19). ACE2 represents the point of entry for SARS-CoV and SARS-CoV-2 within cells [13] through the binding to the surface spike protein of the virus. The almost ubiquitous distribution of ACE2 makes a large number of organs and tissues potentially susceptible to SARS-CoV-2 infection, thus the systemic consequences of COVID19 are well documented, beyond the harmful impact on respiratory function. However, within this context, little information is available on the male reproductive system. The presence of ACE2 has been demonstrated in human seminal plasma [14] and testicular cells, including Leydig cells, Sertoli cells and spermatogonia [15], yet the expression and localization of ACE2 have not yet been investigated in mature human spermatozoa.

In the present study, we demonstrated for the first time the expression of full-length ACE2 and its isoform, short-ACE2, in human spermatozoa at the protein level. We performed western blot analysis of sperm cell lysates using antibodies to ACE2 able to recognize epitopes on different regions of the protein. Immunocytochemistry analysis coupled with confocal microscopy showed the specific location of the proteins on the sperm head and the flagellum. By flow cytometry analysis of viable motile sperm suspensions, we assessed the expression of ACE2 on sperm surface.

## 2. Results and Discussion

### 2.1. ACE2 and Short ACE2 Are Expressed in Human Spermatozoa

Both full-length ACE2 and short ACE2 protein isoforms were found in human sperm lysates after western blot analysis (Figure 1). While there is a clear inter-individual variability, our results showed two specific bands, one of approximately 120 KDa, corresponding to the glycosylated full-length ACE2, and a second one of approximately 52 KDa, recently termed short-ACE2 [16]. As a positive control for western blotting, we used mouse testis, which were positive for both the full-length [5] and the short-ACE2 isoforms (Appendix A).

For immunoblotting analysis, we used two commercial antibodies recognizing different epitopes for the protein: a polyclonal rabbit anti-human (anti-ACE2-1) that recognizes the sequence comprising the amino acids (aas) 750–805 from the C-terminal domain, shared among the two isoforms, and a monoclonal rabbit anti-human antibody (anti-ACE2-2) that recognizes a segment of the zinc metallopeptidase domain corresponding to the aas 200–230 of the N-terminal domain present in the full-length isoform. A third antibody was used in this study, concretely for the flow cytometry experiments. Anti-ACE2-3 corresponds to a polyclonal goat anti-human (anti-ACE2-3) specific to the sequence of aas 18–740 recognizing mainly the full-length isoform. Figure 2 shows the three antibodies used for this study (anti-ACE2-1 and anti-ACE2-2 for western blot and immunocytochemistry, and anti-ACE2-3 for flow cytometry analysis) and their target sequences. While the full-length ACE2 isoform is composed by 805 amino acids, with a length of 92 KDa, the short isoform consists in 459 aas (corresponding to the aas 347–805 of the full isoform) and 52 KDa length (UniProtKB, accession number Q9BYF1; https://www.uniprot.org/uniprot/Q9BYF1; accessed on 4 July 2021). Thus, combining the three antibodies used separately on the same blots after stripping, we were able to validate the specificity of the antibodies and verify the results obtained.

The identification of the short ACE2 isoform has been recently published [16]. This isoform is expressed in human nasal and bronchial respiratory epithelia and upregulated in response to interferon (IFN) treatment or rhinovirus infection, but not after SARS-CoV-2 infection. The short ACE2 isoform was previously regarded as a cleavage product of full ACE2 [17] or a truncated form of the protein (delta ACE2 or dACE2 [18], also termed MIRb-ACE2 [19]). Further studies are warranted to decipher whether the short isoform expressed in mature spermatozoa is a cleavage product from the full-length ACE2 or whether it is originated during the spermatogenesis.

ACE2 has been found in adult Leydig cells, where it might play a regulatory role in steroidogenesis [20]. The evidence presented here constitutes the first demonstration that mature human spermatozoa also express ACE2 at the protein level. Prior to this study, some research groups had reported the mRNA presence in germ cells [21] and in transcriptome sequencing data from human sperm cells [22]; however, these findings did not confirm the expression of the ACE2 protein by mature spermatozoa, as these cells do not present a transcription machinery [23,24]. This notion may have implications in view of the biological role of ACE2 as a gateway for SARS-CoV2 entry within the cell. Since the mature spermatozoon is transcriptionally silent and SARS-CoV-2 is an RNA virus, it is unlikely that the latter could affect sperm biology by replicating within the cell. Nevertheless, the effects related to sperm membrane modifications or the interaction with other receptors or proteins cannot be ruled out. Therefore, we believe that evaluating the ACE2 expression and other co-receptors on mature spermatozoa is the first step in the study of SARS-CoV-2 infection and related impact on the reproductive health; this aspect is, however, beyond the scope of the present study. In this regard, it should be emphasized that although spermatozoa are somehow protected within the testis by the blood-testis barriers, an interaction between the virus and the sperm cells could take place within the female genital tract prior to the fertilization event. In addition, as anticipated, ACE2 is not the only entry point for the virus into the cells. TMPRSS2 is co-expressed with ACE2 in several organs and tissues [25,26,27,28,29,30]. In terms of virus response, it would be necessary to study their co-expression to determine the exact mechanism of action of ACE2. Regardless, the ACE2 expression by the spermatozoa highlights multiple questions related to the viral infection and its physiologic role.

### 2.2. Full-Length ACE2 Is Mainly Expressed in Equatorial and Post-Equatorial Regions of the Sperm Head, While Short ACE2 Is Preferentially Distributed in the Midpiece

By examining the fluorescence patterns after incubation with the different anti-ACE2 antibodies, we found that full-length ACE2 is mainly located in the equatorial and post-equatorial region of the sperm head, whereas the short isoform is preferentially expressed in the midpiece of the sperm flagellum (Figure 3). Our results showed a similar pattern between ACE2 and the protein termed testicular ACE (tACE), previously reported on ejaculated human spermatozoa [31,32]. To identify this isoform (tACE), Köhn et al. (1998) used two different peptides of the ACE sequence [33]. We performed the sequence alignment between the two peptides used by Köhn et al. (1998) and the known sequences for the testis-specific ACE, full-ACE2 and short-ACE2 (https://www.uniprot.org/; accessed on 13 July 2021; identification number P12821-3 for testis-specific ACE, Q9BYF1 for full-ACE2, Q9BYF1-3 for short-ACE2) to determine the sequence identity. Interestingly, peptide 1 shared 23/24 aas with the testis-specific ACE and 8/24 aas with the full ACE2, while peptide 2 shared 10/16 aas with the three isoforms (testis-specific ACE, full-ACE2 and short-ACE2). On this basis, it is possible to assume that testis-specific ACE and ACE2 are colocalized in the same regions of the spermatozoon or that they correspond to the same protein.

### 2.3. Full-Length ACE2 Is Expressed on the Surface of Viable Motile Spermatozoa

As the results produced in immunocytochemistry on fixed and permeabilized cells do not reflect surface expression of the antigen, cytofluorimetric experiments were conducted on suspensions of viable motile spermatozoa to verify whether and to what extent the ACE2 protein was expressed on the sperm membrane. Since anti-ACE2-1 (Abcam, ab15348) and anti-ACE2-2 (Novus, NBP2-67692) antibodies are not recognized for use in cytofluorimetry, the primary polyclonal goat anti-ACE2-3 antibody from R&D Systems (AF933), previously validated and recommended for flow cytometry application, was used for this purpose. Anti-ACE2-3 is a specific antibody for the sequence of aas 18–740, recognizing mainly the full-length isoform (Figure 2).

In twelve experimental settings, semen samples displayed a wide between-subject variability both in the percentage of ACE2-positive spermatozoa (representing a minority of the whole sperm population, ranging from 6.20% to 34.35%) and in the density of protein surface expression, as indicated by the fluorescence peak channel values, ranging from 19 arbitrary units (AU) to 930 AU (Figure 4). It may be hypothesized that the well-known inter-individual variability of seminal parameters [34] could also be reflected in this remarkable width of the expression range of ACE2 in human sperm.

## 3. Materials and Methods

### 3.1. Human Sample Collection

Semen samples were collected by masturbation from 40 healthy normozoospermic non-remunerated volunteers, aged 24–36 years who were not infected with the SARS-CoV-2 and had not been vaccinated at the moment of the semen collection. All subjects signed an informed consent statement. None of the participants had a history of infertility or recognizable risk factors of male infertility. According to recommended procedures of the World Health Organization (WHO) [35], all samples were produced into sterile containers and left for at least 30 min to liquefy before processing. Appendix A shows mean values ± standard error (SE) of standard semen parameters of the 40 donors.

Motile sperm suspensions were obtained by the swim-up procedure. Briefly, spermatozoa were washed twice (700×g for 7 min) in Biggers Whitten Whittingham (BWW) medium. After the second centrifugation, supernatants were removed by aspiration, leaving 0.3 mL on the pellet, and after a 40 min incubation, supernatants containing highly concentrated motile sperm were carefully aspirated. Sperm concentration was evaluated and the cells were washed twice with 100 µL of cold phosphate buffered saline (PBS). Samples were centrifuged at 4 °C for 10 min at top speed and sperm pellets were stored at −80 °C with 1 µL of protease inhibitor cocktail (Merck-Sigma Aldrich, Milan, Italy).

### 3.2. Chemicals

Unless otherwise stated, all the chemicals were purchased from Merck-Sigma Aldrich (Milan, Italy). Three anti-ACE2 antibodies recognizing different epitopes of the protein were used: anti-ACE2-1 rabbit polyclonal (Abcam, Cambridge, MA, USA, ab15348); anti-ACE2-2 rabbit monoclonal (Novus, Toronto, ON, Canada, NBP2-67692); and anti-ACE2-3 goat polyclonal (R&D Systems, McKinley Place, MN, USA, AF933).

### 3.3. Western Blot

Sperm cells were lysed with a RIPA buffer supplemented with 1% halt^TM^ protease and phosphatase inhibitor cocktails (ThermoFisher, St. Louis, MO, USA), sonicated for 15 s at 50% intensity and 4 °C and placed on ice for 30 min gently shaking and vortexing every 10 min. Samples were then centrifuged for 15 min at 15,000× *g* and 4 °C and supernatant containing proteins were then heat-denatured at 100 °C for 5 min in reducing Laemmli sample buffer 4× (BioRad, Philadelphia, PA, USA). Protein content was determined using a MicroBCA™ Protein Assay Kit (ThermoFisher) and read at EnSpire Multimode Plate Reader (PerkinElmer, Waltham, MA, USA). Equal amounts of protein per lane (20 µg) were loaded on an SDS-PAGE 4–15% gradient gel (Mini-PROTEAN^®^ TGX™ Precast Protein Gels, BioRad) and blotted on a nitrocellulose membrane using the Trans-Blot^®^ Turbo^TM^ Transfer System (BioRad). Membranes were blocked in EveryBlot Blocking Buffer (BioRad) for 5 min at room temperature (RT) and then incubated overnight at 4 °C with primary antibodies to ACE2 (1 µg/mL for ab15348, Abcam and 1:1000 for Novus, NBP2-67692, followed by HRP-conjugated secondary antibodies to rabbit IgG (1:10,000, Santa Cruz Biotechnology, Santa Cruz, CA, USA). Bound antibodies were detected using SuperSignal™ West Pico PLUS Chemiluminescent Substrate (ThermoFisher) with the image digitally captured using an Azure C400 (Chemiluminescent Western Blot Imaging System, Azure Biosystems, Dublin, CA, USA). After antibody detection, blots were stripped using a Restore™ Western Blot Stripping Buffer (Thermo Fisher) for 30 min at room temperature and re-incubated following the described protocol. Total labelled protein amount detected by Ponceau S was used as loading control as previously reported (Appendix A) [36]. Blots were cut prior to hybridization, never after. Multiple exposure images with full-length membranes and membrane edges were added to the Appendix A.

### 3.4. Immunocytochemistry

Spermatozoa were fixed and permeabilized with a solution containing 4% paraformaldehyde and 0.5% Tryton X-100 for 20 min at RT. Sperm cells were then washed three times with PBS, centrifuging for 5 min at 500× *g*. The pellet was resuspended and a volume of 20 µL was added to glass slides, inside a 1cm diameter circles. Once the slides were dry, bovine serum albumin (BSA) 1% was added as a blocking agent for the sperm cells for 30 min at RT within a dry chamber. After washing, spermatozoa were incubated overnight at 4 °C with the primary antibody (Abcam ab15348 at 20 µg/mL and Novus NBP2-67692 0.5 µL in 50 µL). After washing three times, a secondary antibody was added as follows: anti-rabbit FITC conjugated for Abcam ab15348 and Novus SN0754, at a dilution 1:250 incubated for 1 h at RT. To stain the nuclei, 4′,6-diamidino-2-phenylindole (DAPI) staining was added during the last 15 min of incubation with the secondary antibodies. Sperm cells were washed again and observed with confocal microscopy (Nikon AR1 laser confocal scanning microscope coupled to the NIS-Element Software, Florence, Italy).

### 3.5. Flow Cytometry

Viable and motile sperm suspensions obtained by swim-up procedure were incubated for 60 min at 37 °C with the primary goat anti-ACE2 antibody (R&D Systems, AF933) (0.25 µg/10^6^ cells). A non-specific serum from a non-immunized goat, at the same concentration of the primary antibody, was used as a negative control. After washing three times with PBS, spermatozoa were incubated with rabbit anti-goat Cy3 conjugated (1:1000) for 60 min at 37 °C. After washing three times, samples were analyzed at Becton Dickinson FACSCalibur (Rome, Italy). At least 2 × 10^4^ events were acquired and analyzed in each sample using Cell Quest software (San Jose, CA, USA). The percentage of positive spermatozoa and the peak channel, expressed as AU, were analyzed.

## 4. Conclusions

In conclusion, from the experiments discussed here, it has been demonstrated that full-length ACE2 and short ACE2 proteins are expressed by mature spermatozoa after ejaculation, strongly confirmed with the three-antibody strategy coupled with immunofluorescence and cytofluorimetry analysis. The demonstration of receptor protein expression represents a key premise for studying the possible impact of SARS-CoV-2 infection on sperm biology.

## Figures and Tables

**Figure 1 ijms-23-03694-f001:**
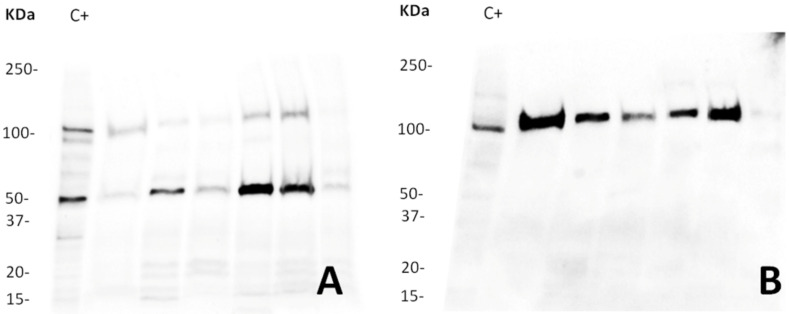
Western blot analysis of ACE2. The image illustrates (**A**) the glycosylated full-length ACE2 (120 KDa) and short-ACE2 isoforms (52 KDa) recognized with anti-ACE2-1 and (**B**) the glycosylated full-length ACE2 (120 KDa) detected by anti-ACE2-2. Antibodies were incubated on the same blot after membrane stripping and re-blotting. Blots were cut prior to hybridization. Each of the six lanes contains 20 µg of protein from semen of different donors. At least four independent experiments with different donors were performed. C+: protein from mouse testis.

**Figure 2 ijms-23-03694-f002:**
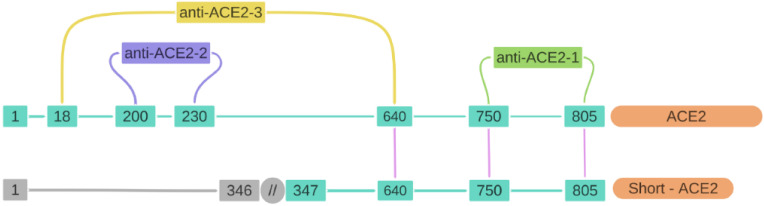
Representative schema of the ACE2 and short-ACE2 sequences and anti-ACE2 antibodies. Full-length ACE2 is composed of 805 amino acids (aas), while short-ACE2 is 459 aas length, thus sharing the sequence between the aas 347–805 (including the C-terminal domain). Numbers in blue correspond to the aas present in each sequence, while numbers in grey correspond to the sequence of aas missing in the short-ACE2 isoform. Three different antibodies were used: anti-ACE2-1 (abcam, ab15348) that recognizes the sequence between the aas 750–805; anti-ACE2-2 (Novus, NBP2-67692), recognizing the aas 200–230 from the full-ACE2; and anti-ACE2-3 (AF933, R&D Systems) recognizing the aas 18–640, mainly present in the full-ACE2 isoform.

**Figure 3 ijms-23-03694-f003:**
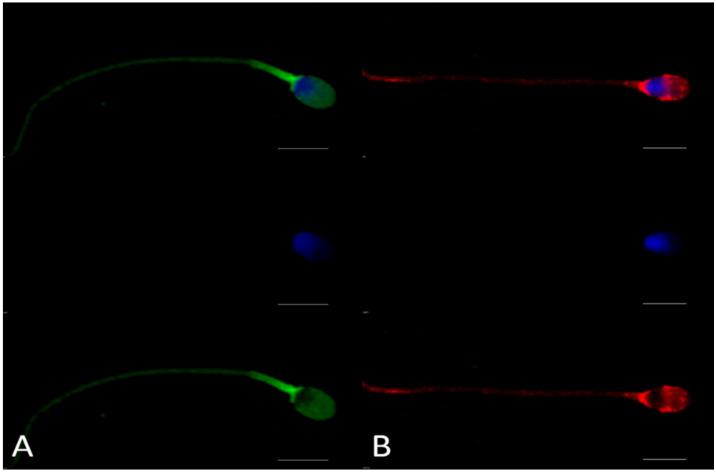
Immunocytochemistry analysis of ACE2 on ejaculated human sperm cells. Panel (**A**) shows the location of the C-terminal domain of ACE2 recognized by anti-ACE2-1 (Abcam, ab15348), shared between both isoforms; Panel (**B**) shows the fluorescence pattern after immunocytochemistry analysis with anti-ACE2-2 (Novus, NBP2-67692), recognizing the full-length ACE2 only. DAPI was used to stain the nuclei. Top: two-laser image (anti-ACE2 antibody + DAPI); middle: sperm cells stained only with DAPI; bottom: sperm cells illustrating only ACE2 fluorescence patterns (scale bar = 5 µm). Negative controls for immunocytochemistry assays are shown in Appendix A.

**Figure 4 ijms-23-03694-f004:**

Surface expression of ACE2 in viable motile human spermatozoa selected by the swim-up procedure as evaluated by flow cytometry. Typical histograms of fluorescence intensity (FL2-H) in spermatozoa incubated with (**A**) a non-specific serum from non-immunized goat (negative control) or (**B**,**C**) two representative sperm samples incubated with primary goat antibody against human ACE2 (anti-ACE2-3 polyclonal antibody, R&D Systems, AF933).

## Data Availability

The data presented in this study are available within the text, in the Appendix A. Further data can be provided upon request to the corresponding author.

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
