# Peer review of "ACE2 Receptor and Its Isoform Short-ACE2 Are Expressed on Human Spermatozoa"

_ijms, 2022, doi:10.3390/ijms23073694_

Round 1
Reviewer 1 Report
The authors considered the expression pattern of ACE2 receptor and its isoform short-ACE2 in human spermatozoa. This is interesting article; however, the following questions should be clarified before publication:
Minor comments
- There are some grammatical errors within the manuscript which need to be revised in right order.
- It would be well if you could write some of main findings as quantitative or mean ±SD within the abstract/result.
- Error bars in bar and linear charts should be explained whether they are standard error bars (SE) or standard deviation (SD)?
Major comments
- It would be well if the authors create a table summarizing the quality of spermatozoa (e.g. sperm counts, morphology, motility, etc.) based on basic standard semen analysis.
- Why you didn’t enter another group as infertile subjects? It would be well if you have a comparison between fertile and infertile regarding the expression of ACE2.
- Why you didn’t provide a quantitative analysis for ACE2 in spermatozoa of the both fertile and infertile men? It would strengthen your article if you could measure ACE2 and evaluate its relationship with other parameters such as age, sperm parameters quality, fertility potential, etc.
- To confirm the importance of ACE2 expression in sperm motility, quantitative analysis and its correlation to sperm motility is essential.
Author Response
We thank the Editor in Chief and the Reviewers for their time, suggestions and comments that we have carefully considered to improve the quality of this manuscript.
This Response file is accompanied by two manuscript files: ACE2_clean version (definitive version) and ACE2_highlighted modifications (all the changes were highlighted). The English language has been reviewed and a Supplementary table with the semen parameters has been added to the supplementary material.
We hope that thanks to the extensive revision and the inclusion of the missing information, our manuscript could be re-considered for publication in International Journal of Molecular Sciences.
Reviewer 1
The authors considered the expression pattern of ACE2 receptor and its isoform short-ACE2 in human spermatozoa. This is interesting article; however, the following questions should be clarified before publication:
Minor comments
- There are some grammatical errors within the manuscript which need to be revised in right order.
- It would be well if you could write some of main findings as quantitative or mean ±SD within the abstract/result.
- Error bars in bar and linear charts should be explained whether they are standard error bars (SE) or standard deviation (SD)?
Reply: We thank R1 for the time implied in reviewing our manuscript. We have revised the whole manuscript and corrected the grammatical errors (all the changes were highlighted in the highlighted version). Regarding the statistical comments, in the original version of the manuscript, we did not perform any statistical analysis, thus we did not include mean + SD or SE. Bars in immunofluorescence images refer to scale bars (µm). Following one of your major criticisms, in the revised manuscript we included a supplementary table 1 showing mean ± standard error of semen parameters of the whole study population.
Major comments
- It would be well if the authors create a table summarizing the quality of spermatozoa (e.g. sperm counts, morphology, motility, etc.) based on basic standard semen analysis.
- Why you didn’t enter another group as infertile subjects? It would be well if you have a comparison between fertile and infertile regarding the expression of ACE2.
- Why you didn’t provide a quantitative analysis for ACE2 in spermatozoa of the both fertile and infertile men? It would strengthen your article if you could measure ACE2 and evaluate its relationship with other parameters such as age, sperm parameters quality, fertility potential, etc.
- To confirm the importance of ACE2 expression in sperm motility, quantitative analysis and its correlation to sperm motility is essential.
Reply: We thank the R1 for these very interesting suggestions. A total of 40 normozoospermic donors were enrolled in our study. To better highlight this aspect, following your comment, we have added a table (supplementary table 1) from which it can be observed that the mean values of all standard parameters of the seminal analysis is above the 25th percentile of the distribution exhibited by the fertile male population (WHO, 2010). Donors were enrolled based on the following inclusion criteria: normal semen quality, absence of medical diseases in their history, absence of previous SARS-CoV2 infection and/or vaccination. Although seminal values above the 25th percentile are known to be associated with full fertilizing capacity (Bonde JP, Lancet 1998;352(9135):1172-7), history of infertility (failure to conceive after 12 months of targeted unprotected intercourse) was not specifically investigated in the present study. Consequently, it is not possible to perform comparative analyses by categorizing the study population according to fertility/infertility status. This could represent an interesting insight for future studies designed for this purpose. Regarding point 4, the stated purpose of the present study was not to explore the relationship between ACE2 expression and sperm functions but only to demonstrate the presence and localization of ACE2 expression in human spermatozoa. Based on the current state of knowledge, any hypothesis regarding the involvement of ACE2 in the regulation/deregulation of biological activities of the spermatozoon would be speculative.
Reviewer 2 Report
In this manuscript, Ramal-Sanchez et al. strive to ellucidate the presence and/or localization of the ACE2 receptor and its isoform in human spermatozoa. Since as appropriately stated by the authors, this receptor plays a pivotal role in the cellular interaction with SARS-CoV-2, a precise description of ACE2 in ejaculated spermatozoa could have significant implications in future management strategies of COVID-associated male subfertility. In this sense, I consider the manuscript novel and highly up-to-date. The experimental approach takes advantage of adequate methods, and the manuscript reads well. Nevertheless, I do have some questions for further clarification and perhaps a few points open for discussion:
- It is not clear how many donors were involved in the study. Although the authors state that none of the donors was vacccinated, it was not clear if these subjects had had COVID prior to the involvement in the study. Perhaps it would be interesting to divide the subjects into 2 groups depending on whether they had tested positive for COVID previously or not. This aspect could also be explored in further studies.
- What were the inclusion/exclusion criteria following the WHO guidelines? Did the authors evaluate the sperm motility, viability or morphology prior to including the samples into further experiments?
- Although the authors stste that this study provides preliminary data, I am curious about any speculations as to possible implications of the presence of ACEs in the sperm midpiece and/or postacrosomal region of the spemr head. Is it possible that an active SARS-CoV-2 infection could have implications of the mitochondrial metabolism or sperm morphology for example?
- What would be the limitations of the study?
- Based on what criterion were specific samples selected for Western blot? Were these chosen randomly or based on any factor/property of the sample?
- What would explain an obvious variability among the samples as observed in the Western blot/flow cytometric analysis?
Author Response
We thank the Editor in Chief and the Reviewers for their time, suggestions and comments that we have carefully considered to improve the quality of this manuscript.
This Response file is accompanied by two manuscript files: ACE2_clean version (definitive version) and ACE2_highlighted modifications (all the changes were highlighted). The English language has been reviewed and a Supplementary table with the semen parameters has been added to the supplementary material.
We hope that thanks to the extensive revision and the inclusion of the missing information, our manuscript could be re-considered for publication in International Journal of Molecular Sciences.
In this manuscript, Ramal-Sanchez et al. strive to ellucidate the presence and/or localization of the ACE2 receptor and its isoform in human spermatozoa. Since as appropriately stated by the authors, this receptor plays a pivotal role in the cellular interaction with SARS-CoV-2, a precise description of ACE2 in ejaculated spermatozoa could have significant implications in future management strategies of COVID-associated male subfertility. In this sense, I consider the manuscript novel and highly up-to-date. The experimental approach takes advantage of adequate methods, and the manuscript reads well. Nevertheless, I do have some questions for further clarification and perhaps a few points open for discussion:
Reply: We thank R2 for his/her time implied in reviewing our manuscript, as well as for the interesting comments, suggestions and questions open for discussion.
- It is not clear how many donors were involved in the study. Although the authors state that none of the donors was vacccinated, it was not clear if these subjects had had COVID prior to the involvement in the study. Perhaps it would be interesting to divide the subjects into 2 groups depending on whether they had tested positive for COVID previously or not. This aspect could also be explored in further studies.
- What were the inclusion/exclusion criteria following the WHO guidelines? Did the authors evaluate the sperm motility, viability or morphology prior to including the samples into further experiments?
Reply: A total of 40 normozoospermic donors were enrolled in our study. To better highlight this aspect, following your comment, we have added a table (supplementary table 1) from which it can be seen that the mean values of all standard parameters of the seminal analysis is above the 25th percentile of the distribution exhibited by the fertile male population (WHO 2010). As better stated in the revised manuscript (see page 5, lines 197-200), donors were enrolled based on the following inclusion criteria: normal semen quality, absence of medical diseases in history, absence of previous SARS-CoV2 infection and/or vaccination.
- Although the authors stste that this study provides preliminary data, I am curious about any speculations as to possible implications of the presence of ACEs in the sperm midpiece and/or postacrosomal region of the spemr head. Is it possible that an active SARS-CoV-2 infection could have implications of the mitochondrial metabolism or sperm morphology for example?
- What would be the limitations of the study?
Reply: in our humble opinion, this is a very interesting question, and we are curious too. However, we don’t have an answer yet, especially considering the large between-subjects variability. In this light, it remains to be elucidated not only whether ACE2, in its different isoforms, plays a role in sperm physiology but also whether and to what extent the SARS-CoV2/ACE2 interaction affects sperm functions. Indeed, taking into account that spermatozoa are protected by the blood-testis barrier within the testis, it is hard to hypothesize how and where do they get in contact with the virus (in the rete testis, seminiferous tubules, within female genital tract?). On the other hand, we could hypothesize not only a direct but also an indirect effect derived from the SARS-CoV-2 infection, which has recently been observed to damage some tissues but also to trigger a cytokines response that might in turn affect both testis and sperm. The absence of functional assessments may indeed be considered a limitation of our research, however these analyses would have been beyond the stated aims of the study.
- Based on what criterion were specific samples selected for Western blot? Were these chosen randomly or based on any factor/property of the sample?
Reply: Samples were chosen randomly among the 40 available normozoospermic ejaculates, and two technical replicates were performed with each sample.
- What would explain an obvious variability among the samples as observed in the Western blot/flow cytometric analysis?
Reply: We consider that the cause could be the well-known inter-individual variability of the seminal parameters. Following your comment, we added the following statement in the revised manuscript: “It may be hypothesized that the well-known inter-individual variability of seminal parameters [34] could also be reflected in this remarkable width of the expression range of ACE2 in human sperm” (page 5, lines: 185-187).